

# Development of a novel prediction model based on protein structure for identifying RPE65-associated inherited retinal disease (IRDs) of missense variants

Jiawen Wu[1,*], Zhongmou Sun[2,*], Dao wei Zhang[1], Hong-Li Liu[1], Ting Li[1], Shenghai Zhang[1,3,4,5] and Jihong Wu[1,3,4,5]

[1] Eye Institute, Eye and ENT Hospital, College of Medicine, Fudan University, Shanghai, China
[2] University of Rochester School of Medicine and Dentistry, New York, United States of America
[3] Shanghai Key Laboratory of Visual Impairment and Restoration, Science and Technology Commission of Shanghai Municipality, Shanghai, China
[4] State Key Laboratory of Medical Neurobiology, Institutes of Brain Science and Collaborative Innovation Center for Brain Science, Shanghai, China
[5] Key Laboratory of Myopia, Ministry of Health, Shanghai, China
[*] These authors contributed equally to this work.

Corresponding author
Jihong Wu, jihongwu@fudan.edu.cn

## ABSTRACT

**Purpose.** This study aimed to develop a prediction model to classify *RPE65*-mediated inherited retinal disease (IRDs) based on protein secondary structure and to analyze phenotype-protein structure correlations of *RPE65* missense variants in a Chinese cohort.

**Methods.** Pathogenic or likely pathogenic missense variants of *RPE65* were obtained from UniProt, ClinVar, and HGMD databases. The three-dimensional structure of RPE65 was retrieved from the Protein Data Bank (PDB) and modified with Pymol software. A novel prediction model was developed using LASSO regression and multivariate logistic regression to identify *RPE65*-associated IRDs. A total of 21 Chinese probands with *RPE65* variants were collected to analyze phenotype-protein structure correlations of RPE65 missense variants.

**Results.** The study found that both pathogenic and population missense variants were associated with structural features of RPE65. Pathogenic variants were linked to sheet, $\beta$-sheet, strands, $\beta$-hairpins, $Fe^{2+}$ (iron center), and active site cavity, while population variants were related to helix, loop, helices, and helix–helix interactions. The novel prediction model showed accuracy and confidence in predicting the disease type of *RPE65* variants (AUC = 0.7531). The study identified 25 missense variants in Chinese patients, accounting for 72.4% of total mutations. A significant correlation was observed between clinical characteristics of RPE65-associated IRDs and changes in amino acid type, specifically for missense variants of F8 (H68Y, P419S).

**Conclusion.** The study developed a novel prediction model based on the protein structure of RPE65 and investigated phenotype-protein structure correlations of RPE65 missense variants in a Chinese cohort. The findings provide insights into the precise diagnosis of *RPE65*-mutated IRDs.

Retinitis pigmentosa(RP), Inherited retinal disease (IRDs), Phenotype, Machine learning

## INTRODUCTION

The retinal pigment epithelium-specific 65 kD protein (RPE65) is a retinoid isomerohydrolase that plays a critical role in the regeneration of 11-cis retinol in the visual cycle. Encoded by the *RPE65* gene (OMIM 180069), this protein is expressed exclusively in the retinal pigment epithelium (*Moiseyev et al., 2005*). Autosomal recessive mutations in RPE65, often involving bi- or multi-allelic mutations, can lead to photoreceptor degeneration in humans (*Gu et al., 1997*). Clinically, a majority of *RPE65* variants (approximately 67%) are commonly associated with Leber congenital amaurosis (LCA) or retinitis pigmentosa (RP) phenotypes (approximately 16%) (*Stenson et al., 2017*), which have similar fundus manifestations (*Aoun et al., 2021*). In 2017, the US Food and Drug Administration approved voretigene neparvovec (Luxturna®; Spark Therapeutics, Philadelphia, PA, USA) gene therapy for the treatment of patients with viable retinal cells and confirmed biallelic *RPE65* mutation-associated retinal dystrophy (*Russell et al., 2017*). However, while timing the initiation of gene therapy is an important consideration, we still need more information about the natural history of the disease to better guide clinical applications (*Botto et al., 2022*; *Sodi et al., 2021*). Therefore, understanding the correlation between mutation and phenotype is critical.

RPE65 is a beta-propeller fold protein comprised of seven blades. Splicing and frameshift mutations in this protein can result in a truncated and non-functional protein product that is presumed to be null and irrelevant (*Gu et al., 1997*). However, predicting the significance of missense variations associated with *RPE65*-mediated inherited retinal diseases (IRDs) is challenging as both benign and pathogenic variations coexist in almost every disease-associated gene (*Lek et al., 2016*). In *RPE65*-associated IRDs, the analysis of missense mutants is particularly challenging as it is difficult to predict the significance of variants of uncertain significance (VUS). Mutations causing disease often occur in regions with secondary protein structures, which are crucial for protein stability and function (*Khan & Vihinen, 2007*; *Yue, Li & Moult, 2005*). In 2009, the crystal structure of RPE65 was resolved, revealing details of its active site architecture and oligomeric state (*Kiser et al., 2009*). Recent research has identified notable features of the RPE65 protein structure, including an iron center coordinated by a 4-His/3-Glu motif, a hydrophobic/cationic patch on the protein's exterior, an active site cavity, and a dimeric form (*Kiser, 2022*). Despite the knowledge that protein secondary structure influences protein function, few studies have investigated the correlations between secondary structure and missense/phenotypes in *RPE65*-mediated IRDs (*Lorenz et al., 2008*; *Thompson et al., 2000*).

In this study, we aimed to investigate the relationship between missense variations in *RPE65* and their impact on protein features. We analyzed missense variants in both pathogenic and normal populations and developed a new prediction model based on protein structure to calculate the risk of VUS occurring in *RPE65* missense variants. We

also examined the correlations between secondary structure and phenotypes in probands with *RPE65* variants to gain new insights into the role of *RPE65* missense variants in the pathogenic mechanisms of *RPE65*-associated IRDs. Our study provides important guidance for future gene therapy strategies and could ultimately lead to more effective treatment options for *RPE65*-associated IRDs.

## MATERIALS & METHODS

### Data collection

Missense variants of the pathogenic (P), likely pathogenic (LP), and variants of uncertain significance (VUS) of *RPE65* were downloaded from the National Library of Medicine database (ClinVar, https://www.ncbi.nlm.nih.gov/clinvar/?term=RPE65%5Bgene%5D) and Human Gene Mutation Database (HGMD, https://www.hgmd.cf.ac.uk/ac/index.php). Missense variants of the study population were downloaded from the UniProt database (https://www.uniprot.org/uniprot/Q16518#expression).

### Subject

The current study was approved by the Ethics Committee of the Eye and ENT Hospital of Fudan University and conformed to the tenets of the Declaration of Helsinki (2018021). Written informed consent was obtained from all participants or their guardians. 21 Chinese probands were enrolled from July 2018 to March 2022. All of the clinical examinations were performed by practiced ophthalmologists, and the patients' medical histories were recorded. In total, RPs patients with *RPE65* variants and their related family members were enrolled in this retrospective analysis if they had the following qualifications: (1) a confirmed diagnosis of RP or LCA clinically; (2) compound heterozygous and homozygous pathogenic or likely pathogenic RPE65 variants that could be related to the phenotype; (3) no other gene mutations. Patients with the following conditions were excluded from the study: (1) other coexisting ocular diseases; (2) a history of trauma or surgery in either eye; (3) complications including epiretinal membranes, retinal detachment, and maculopathy.

### Ophthalmic examination

The ophthalmic examinations conducted included visual acuity testing, slit-lamp biomicroscopy, fundus examination, visual field (VF, Humphrey Visual Field Analyzer, Carl Zeiss, Dublin, California, USA), and full-field electroretinography (ERG). These examinations were conducted according to the standards of the International Society for Clinical Electrophysiology of Vision. Much optic coherence tomography (OCT) data was missing, and therefore not included in this study. Clinical diagnosis of RP and LCA was majorly based on history and ocular examination.

### Molecular analysis

Molecular testing was performed after extracting genomic DNA from the peripheral blood using a custom-designed panel (described in our earlier publication; *Gao et al., 2019*) or whole-exome sequencing (WES). Exonic and adjacent intronic sequences were captured and enriched from genomic DNA using the Roche KAPA HyperExome Chip and were run

on a MgISEQ-2000 sequencer to test mutations. The quality control index of sequencing data with an average sequencing depth in the target area was ≥180X, and the proportion of loci with an average sequencing depth >20X in the target area was >95%. The *RPE65* variant was confirmed by Sanger sequencing.

## Structural biochemistry classification of *RPE65*

The RPE65 protein structure was downloaded from the PDB database (4f3d, https://www.pdbus.org/) and predicted by AlphaFold (*Jumper et al., 2021*). Combined with the sequence obtained from molecular analysis, the RPE65 monomer of humans was modified by Pymol software. Missense mutations were mapped on the RPE65 protein structure. The degree of conservation of the amino acid substitution was assessed using a substitution matrix (BLOSUM 62) (*Stone, 2003*). The RPE65 protein structure was classified using the following features. Firstly, amino acids were classified into four groups based on their physicochemical properties: (1) Non-polar amino acid (NPA): Ala, Leu, Met, Phe, Pro, Tyr, Trp, Ile, Val; (2) Polar neutral amino acid (PNA): Asn, Cys, Gln, Ser, Thr, Gly; (3) Polar basic amino acid (PBA): Arg, Lys, His; (4) Polar acid amino acid (PAA): Asp, Glu (*Lazar et al., 2022*). Missense variants were classified by analyzing whether their physicochemical properties have changed, which is called an amino acid (AA) change. Secondly, the RPE65 structure was classified using the classic 3-class secondary structure: helix, sheet, and loop. Thirdly, the RPE65 structure was identified into 8 motifs, called a PROMOTIF program according to *Hutchinson & Thornton (1996)* (https://www.ebi.ac.uk/thornton-srv/databases/cgi-bin/pdbsum/GetPage.pl?pdbcode=index.html). Finally, the RPE65 structure was classified into four functional features as described by *Kiser et al. (2009)* (Fig. S1). The location information of each amino acid is shown in Table S1. The structure was classified into 16 three-dimensional (3D) features with some overlaps, and every single amino acid site was turned into a digital fingerprint.

## Development of a prediction model

The least absolute shrinkage and selection operator (LASSO) method was used to select the best features of predictive risk factors, and logistic regression analysis was chosen to develop a prediction model. The odds ratio (OR) and C-index were calculated. All statistical tests were two-tailed and $P < 0.05$ was considered a significant difference. The optimal model was selected to draw a nomogram and a calibration curve. The contact of amino acid residues of variants was calculated with Pymol software.

## Statistical analysis

In order to quantify the burden of pathogenic variation or population variation, a two-sided chi-square test or Fisher's exact test was used. Using the R program (version 4.1.3), it was determined an OR >1 and $p < 0.05$ indicate that a particular 3D feature is characteristic of pathogenic variants. Figures were plotted using the R package "ggplot2". The correlation analysis between clinical characteristics and structural features was completed using Kendall's tau b correlation analysis with a two-tailed test, and a $P$ value less than 0.05 in SPSS was considered significant (version 26.0.0.0).

# RESULTS

## Characteristic 3D features of pathogenic and population missense variants of *RPE65*

According to the ClinVar and HGMD databases, missense mutations account for 67.58% and 54.89%, respectively, of all mutations in pathogenic (P and LP) *RPE65* variants. This indicates that missense mutations are the most significant type of mutation in *RPE65* genes (Fig. 1A). After deduplication, we marked 153 pathogenic (P and LP), and 300 population amino acid variations (as reported in the UniProt database) in the RPE65 monomer (Fig. 1B). We then calculated the substitution matrix score (BLOSUM 62) between the two groups and found that the score of population variants was higher than that of *RPE65*-IRDs ($P < 0.0001$) (Fig. 1C). To systematically identify the 3D features associated with "pathogenic" and "population" variants, we analyzed the 3D sites affected in 153 pathogenic variants (ClinVar and HGMD databases) and 300 general population variants (UniProt database) from *RPE65* genes (Table S2). Among the sixteen features, sheet (OR = 2.55, $P = 0.000$), $\beta$-sheet (OR = 2.64, $P = 0.000$), strands (OR = 2.57, $P = 0.000$), $\beta$-hairpins (OR = 1.78, $P = 0.004$), $Fe^{2+}$ (iron center) (OR = 5.73, $P = 0.001$), and active site cavity (OR = 2.76, $P = 0.006$) were significantly correlated with *RPE65*-associated *IRDs* missense variants. Additonally, helix (OR = 0.37, $P = 0.009$), loop (OR = 0.55, $P = 0.003$), helices (OR = 0.42, $P = 0.11$), and helix-helix interactions (OR = 0.43, $P = 0.046$) were also correlated (Fig. 1D).

## Development of a prediction model based on the RPE65 protein structure

LASSO binary logistic regression was utilized to select the top fourteen 3D variables from the sixteen structures analyzed (Figs. 2A and 2B). A nomogram was subsequently created from these variables (Fig. 2C). The receiver operating characteristic (ROC) curves generated demonstrated strong predictive capability with an area under curve (AUC) value of 0.75131 (Fig. 3A). Calibration curves were generated to evaluate the calibration of the *RPE65*-associated IRDs nomogram (Fig. 3B). Next, the risk of *RPE65* missense variants of uncertain significance (VUS) occurring was calculated using a nomogram (Fig. 4A). The highest risk of VUS occurrence was observed in H241R and R44L, where amino acids at both mutation sites were altered. H241R influenced the $Fe^{2+}$ and active site cavity of functional features. Both missense mutations were located in the $\beta$-sheet and in strands of 8-class-secondary structure(Fig. 4B), which were identified as risk factors for pathogenicity (Fig. 1C). Furthermore, changes in residues were observed. The amino acid site 241 changed from Histidine (H) to Arginine (R), decreasing its connection to Y239 (Fig. 4C). Similarly, the residue at position 44R interacted with H68, F469, T525, and F526. In contrast, when mutated to 44L, the residue was only connected to H68 (Fig. 4D). In summary, these findings may indicate that both missense mutations are pathogenic or likely pathogenic.

## Genotype analyses of patients with *RPE65* variants

Twenty-one probands with *RPE65* variants (13 males and eight females) were enrolled in this study (Fig. S2). Missense mutations accounted for 72.4% of the total mutation types

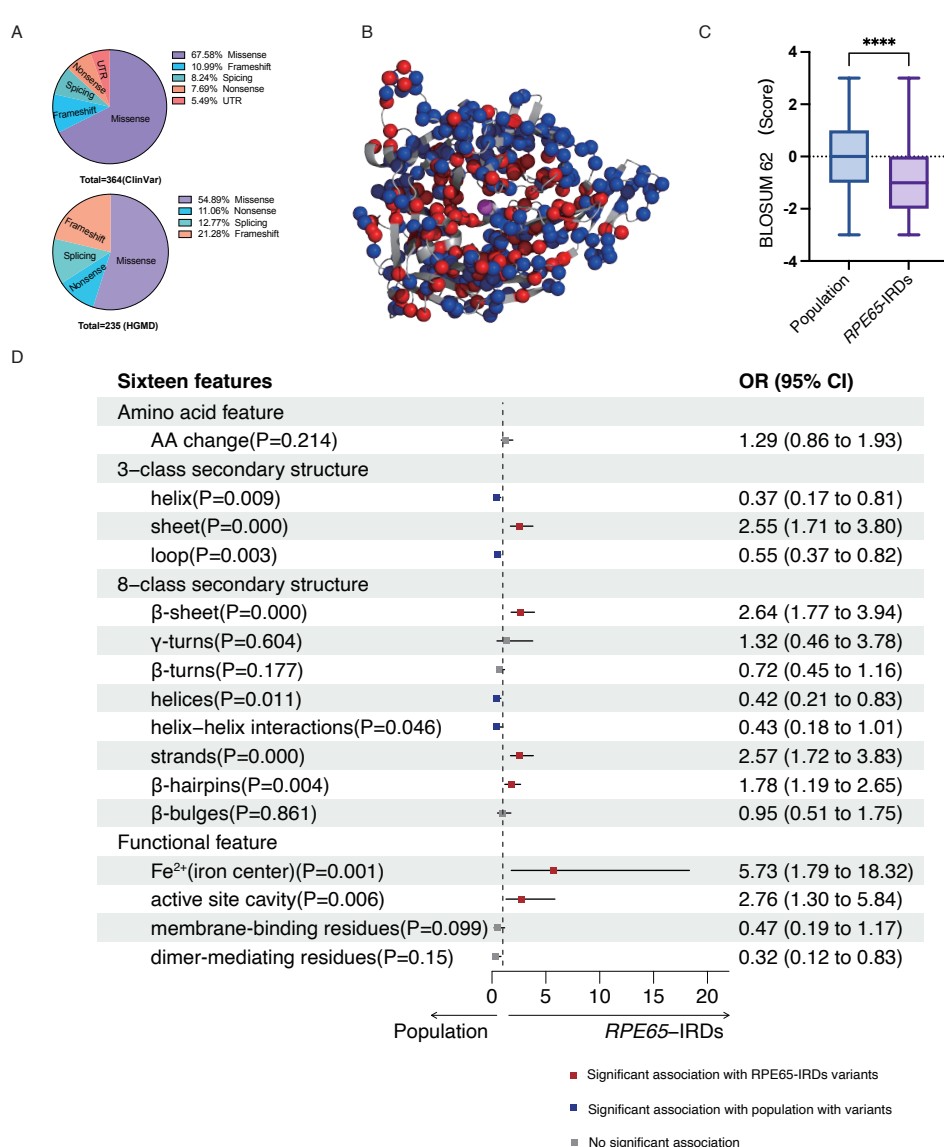

Figure 1 **Association of pathogenic and population missense variations with 16 3D features (a combination of amino acid features, secondary structural, and functional features on protein structure) for RPE65.** (A) The proportion of mutation type of *RPE65* accumulated from the ClinVar database. (B) A total of 153 pathogenic (P and LP), and 300 population amino acid variations overview. Red: RPE65-IRDs missense variants. Blue: population amino acid variations. Magenta: $Fe^{2+}$ (iron center). (C) The box-plot shows the results of the difference between 153 pathogenic (P and LP), and 300 population amino acid variations with BLOSUM 62 score. (D) The plot shows the results of two-sided chi-square test or Fisher's exact tests of association between 153 pathogenic (P and LP), and 300 population amino acid variations with the features. The OR > 1 and OR < 1, along with P < 0.05, indicate that the corresponding feature (*y*-axis) is enriched in pathogenic (red square) and population (blue square) variants, respectively. P, pathogenic; LP, likely pathogenic; CI, confidence interval; OR, odds ratio. **** *P* < 0.0001.

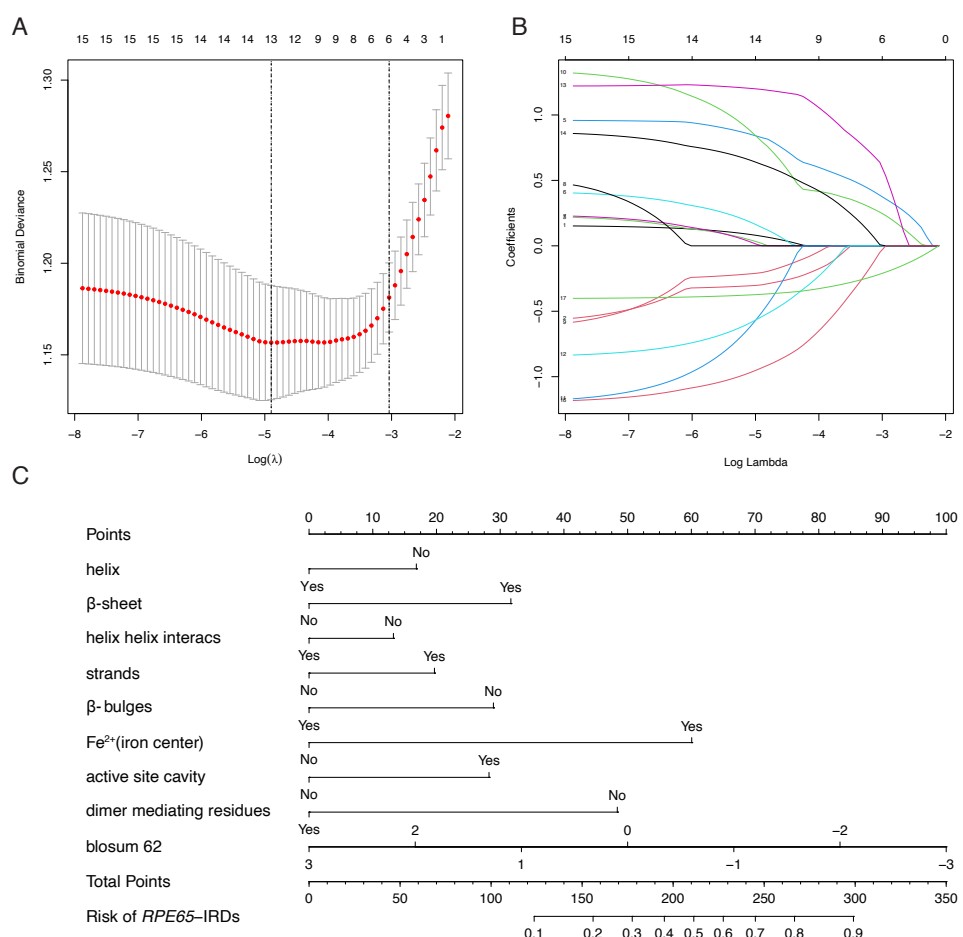

**Figure 2** **Construction of *RPE65*-IRDs variants prediction model.** (A) Optimal parameter (lambda) selection in the LASSO model used fivefold cross-validation *via* minimum criteria. Dotted vertical lines were drawn at the optimal values by using the minimum criteria and the 1 SE of the minimum criteria (the 1-SE criteria). (B) LASSO coefficient profiles of the 17 features. A coefficient profile plot was produced against the log (lambda) sequence. (C) The diagnostic nomogram was developed, with 9 features. LASSO, least absolute shrinkage and selection operator; SE, standard error.

with some overlaps, which were identified in the RPE65 monomer (Fig. 5A). A total of 26 different mutations were observed, including 15 missense, five frameshifts, one nonsense, and five splicing mutations. Thirteen patients had biallelic mutations, and one patient (F13) had four allelic mutations, including one missense (Asp482Asn) and three frameshift mutations (Leu270Hisfs11, Trp271Lysfs11, and Ser269Metfs13) (Table 1). For the purpose of analysis, patient F8 was selected at random (Fig. 5B). The pathogenic missense mutation H68Y (His68Tyr) was located in the sheet, β-sheet, strands, and β-hairpins, all of which are risk factors. In contrast, P419S (Pro419Ser), which is a variant of uncertain significance (VUS), with a BLOSUM 62 score of −1, was located in the sheet, loop, strands, β-hairpins, and β-bulges, and its risk score ranged from 0.3 to 0.4, with no change in residue contact

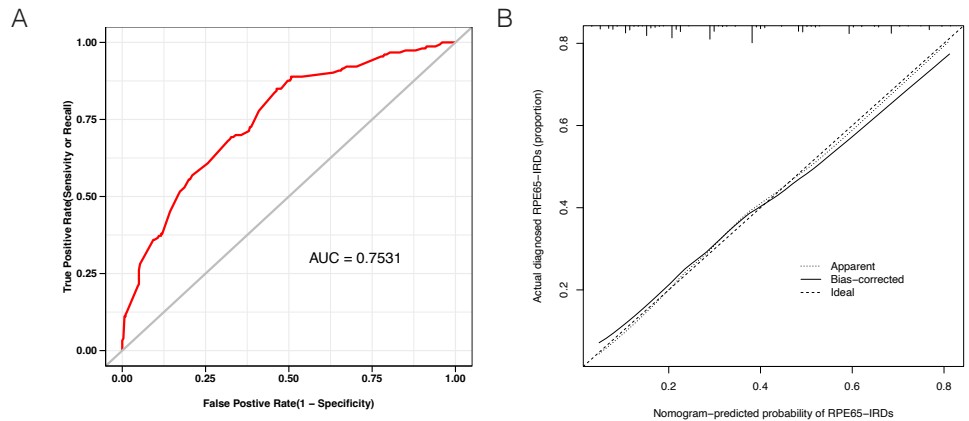

**Figure 3** **Evaluation of the model constructed in this study.** (A) Receiver operating characteristic (ROC) curves showing a good prediction with a 0.7531 value of AUC. (B) The diagonal dotted line represents a perfect prediction by an ideal model. The solid line represents the performance of the nomogram, of which a closer fit to the diagonal dotted line represents a better prediction.

(Fig. 5C). Thus, it can be concluded that P419S has two risk factors and has a high possibility of being pathogenic.

## The correlation between phenotype and protein structure of *RPE65*-mediated IRDs

The correlation between phenotype and protein structure of *RPE65*-mediated IRDs was then analyzed. The clinical characteristics of the patients with *RPE65*-mediated IRDs are presented in detail in Table 2. In this cohort, we conducted a correlation analysis between the phenotype (including BCVA, illness duration, BCVA/illness duration, fundus photography, and ERG) and missense mutations in the protein structure. The results of the analysis are summarized in Table 3, which indicates that BCVA is strongly correlated with an amino acid (AA) change ($R = 0.515$, $P < 0.01$) and $\beta$-hairpins ($R = 0.33$, $P < 0.05$). To eliminate the effects of individual differences, the ratio of BCVA and illness duration was calculated, where a smaller ratio indicates a relatively faster progression of the disease. BCVA/illness duration was found to be correlated with AA change ($R = 0.340$, $P < 0.05$). Furthermore, fundus photography and ERG were found to be correlated with AA change, helix, and helices ($P < 0.01$).

## DISCUSSION

Predicting the 3D structure that a protein will assume based solely on its amino acid sequence has been a significant challenge in research for over 50 years (*Anfinsen, 1973*; *Dill et al., 2008*). A recent study has revealed the molecular effect of missense variants by accomplishing a comprehensive characterization of amino acid positions in protein structures, providing reference for the clinical interpretation of pathogenic and benign missense variants (*Iqbal et al., 2020*). Current studies show that most of the known mutations in the functional region of the RPE65 protein can cause retinal disease (*Kiser,*

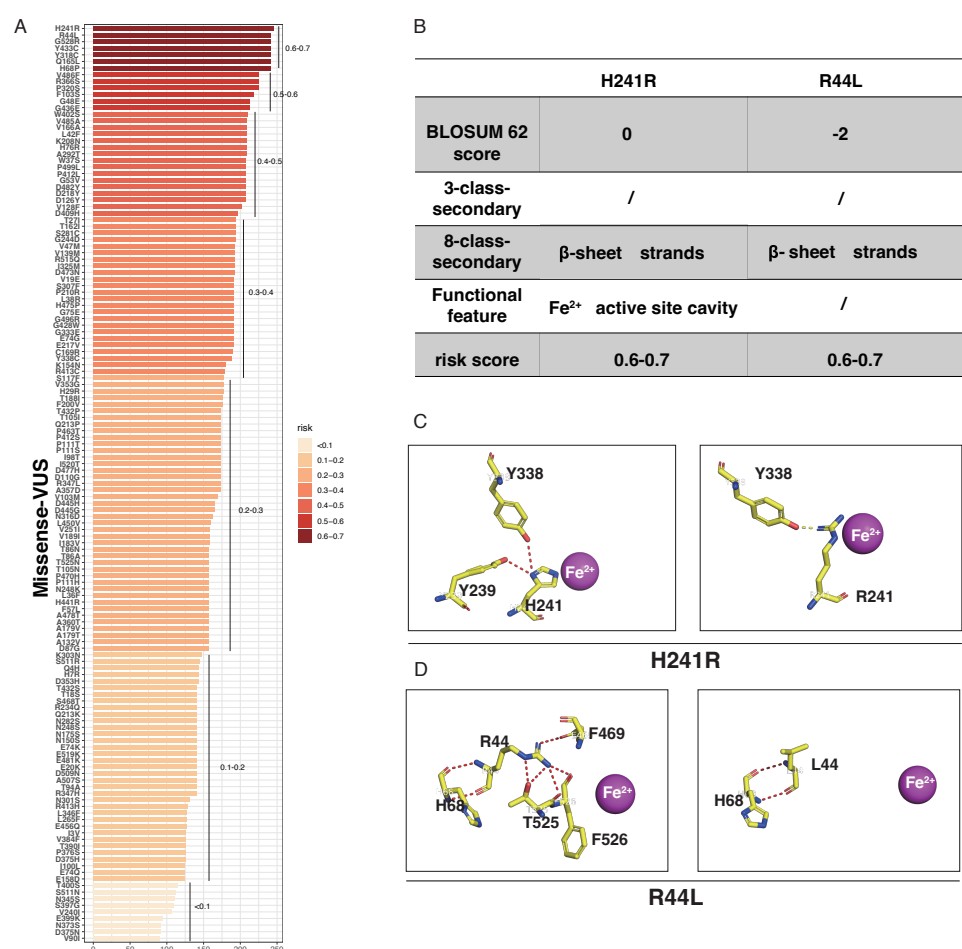

**Figure 4 The perdition of *RPE65* missense of VUS.** (A) Risk prediction of RPE65 missense in a variant of uncertain significance (VUS) according to the nomogram. (B) The RPE65 structure changes with the top highest risk score. (C) Molecular structure binding changes of H241R. (D) Molecular structure binding changes of R44L.

*2022*). However, the way in which the protein structure influences disease remains unclear. In this study, we explored the correlations between the pathogenicity and population of *RPE65* missense variants. Using a relatively large patient sample, we analyzed the association between pathogenic and population missense variations to *RPE65*. We found that the missense variants of *RPE65*-associated IRDs are related to the sheet and $\beta$-sheet, strands, $\beta$-hairpins, iron center, and active site cavity, which may indicate that the missense variants located in these sites are prone to be pathogenic.

In previous studies, researchers focused on clarifying the frequency and phenotypes characteristic of different races or distributions (*Gao et al., 2021*; *Li et al., 2020*; *Lopez-Rodriguez et al., 2021*). However, to date, genotype-phenotype correlations in patients with *RPE65* variants have not been well established (*Gao et al., 2021*). Recently, a study analyzed the molecular characterization of the *RPE65* cohort and genotype-phenotype

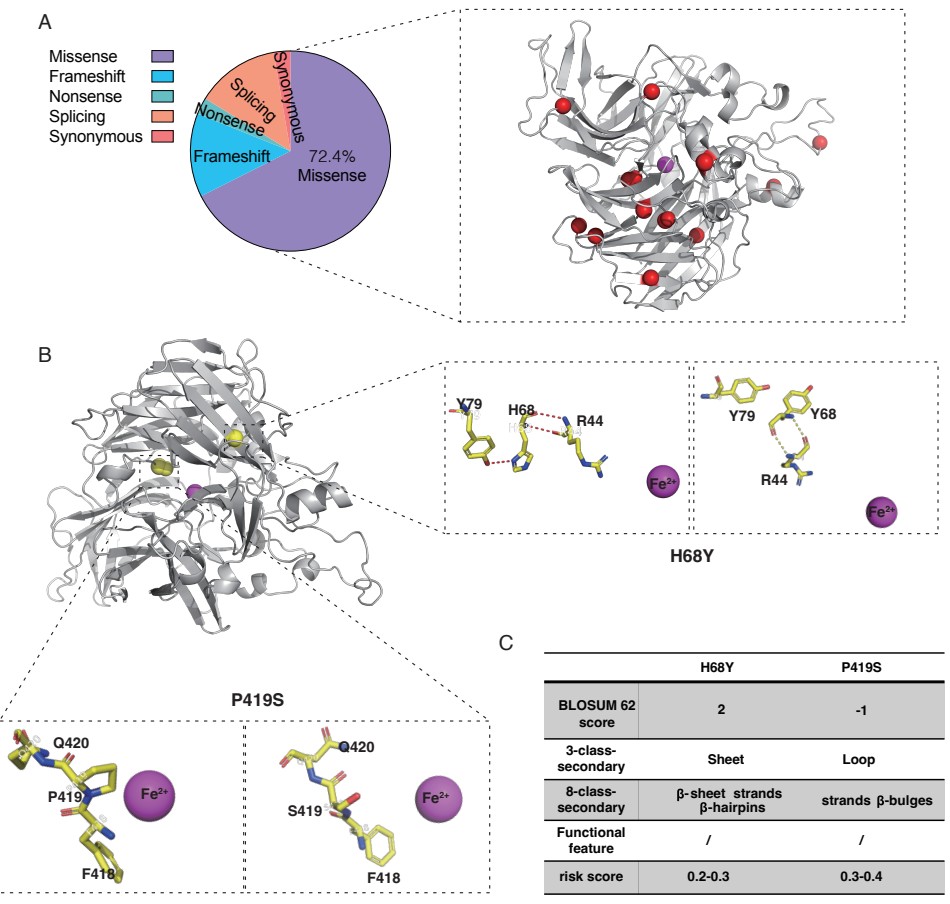

**Figure 5   Characteristic of RPE65 missense in our cohort.** (A) The proportion of RPE65 missense in our cohort. The red balls represented the missense in this cohort, accounting for 72.4% of all mutations. (B) An example of RPE65 missense patients (F8) with biallelic mutations showing amino acid contact changes. (C) The RPE65 structure changes are calculated with a risk score.

correlation according to the number of RPE65 loss-of-function (LoF) alleles in the Italian population (*Testa et al., 2022*). Yet there are no studies analyzing the correlation between phenotype and missense variants. In this study, we indicated that the clinical characteristic was correlated to changes in amino acid residues, helix, helices, and $\beta$-hairpins in the missense mutation of *RPE65*. An AA change may be one of the factors influencing the phenotype of *RPE65*-associated IRDs with missense variants.

Determining whether a genetic variation in a patient is responsible for their disease can be challenging. Previously, researchers attempted to predict the pathogenicity of *RPE65* mutations using an empirical algorithm to estimate pathogenic probability (EPP), which was validated for certain *RPE65* variants (*Philp et al., 2009*; *Stone, 2003*). However, this method may have limited application in many cases. Recently, *Cho et al. (2022)* analyzed carrier frequency and expected incidence of RPE65-associated IRDs in East Asians and Koreans using exome data from the Genome Aggregation Database (gnomAD) and the Korean Reference Genome Database (KRGDB). In our study, we investigated the

**Table 1  RPE65 variants identified in this cohort of patients.**

| Nucleotide change | Amino acid change | Exon/intron | ACMG category | Patients | Mutation type | Reference |
|---|---|---|---|---|---|---|
| c.1399C >G | Pro467Ala | E13 | VUS | F1, F3 | Missense | Reported |
| c.272G >A | Arg91Gln | E4 | P | F2, F15, F21 | Missense | Reported |
| c.271C >T | Arg91Trp | E4 | P | F2, F12 | Missense | Reported |
| c.1338G >T | Arg446Ser | E12 | P | F5, F7 | Missense | Reported |
| c.1543C >T | Arg515Trp | E14 | P | F6, F14, F20 | Missense | Reported |
| c.1444G >A | Asp482Asn | E13 | LP | F6, F13 | Missense | Reported |
| c.1255C >T | Pro419Ser | E12 | VUS | F8 | Missense | Reported |
| c.202C >T | His68Tyr | E3 | P | F8 | Missense | Reported |
| c.1590C >A | Phe530Leu | E14 | LP | F9, F16, F17 | Missense | Reported |
| c.997G >C | Gly333Arg | E9 | VUS | F10 | Missense | Reported |
| c.334T >A | Cys112Ser | E4 | VUS | F10 | Missense | Reported |
| c.93A >G | Thr31Thr | E2 | VUS | F15, F21 | Synonymous | Reported |
| c.335G >A | Cys112Tyr | E4 | VUS | F16 | Missense | Reported |
| c.1520C >T | Ala507Val | E14 | VUS | F17 | Missense | Novel |
| c.1051G >A | Glu351Lys | E10 | VUS | F19 | Missense | Novel |
| c.493C >T | Gln165* | E5 | P | F4 | Nonsense | Reported |
| c.837del | Phe279Leufs46 | E8 | LP | F11 | Frameshift | Novel |
| c.376del | Val126*fs1 | E5 | LP | F12 | Frameshift | Novel |
| c.808_809insA | Leu270Hisfs11 | E8 | LP | F13 | Frameshift | Novel |
| c.809_810insGAAG | Trp271Lysfs11 | E8 | LP | F13 | Frameshift | Novel |
| c.805_806insTGGA | Ser269Metfs13 | E8 | LP | F13 | Frameshift | Novel |
| c.94+2T >A | _ | I2 | LP | F1 | Splicing | Novel |
| c.245+4A >G | _ | I3 | LP | F18 | Splicing | Novel |
| c.354-2A >G | _ | I5 | LP | F7 | Splicing | Novel |
| c.998+1G >A | _ | I10 | LP | F3 | Splicing | Reported |
| c.858+1delG | _ | I9 | LP | F9 | Splicing | Novel |

**Notes.**

E, Exon; I, Intron; P, Pathogenic; LP, Likely pathogenic; VUS, variants of uncertain significance; ACMG, The American College of Medical Genetics and Genomics.

association between populations and the pathogenicity of *RPE65* missense variants and developed a prediction model specifically tailored for missense variants of *RPE65* based on its secondary structure and functional features, without considering other missing factors. We also discussed how to apply this model in clinical practice. For patients whose *RPE65* variants are classified as variants of uncertain significance (VUS), we recommend locating the missense variant and referring to the nomogram to calculate the risk score. If the variant affects structural features strongly correlated with pathogenicity and receives a high-risk score, it is reasonable to classify it as pathogenic. But for variants with limited risk factors or low-risk scores, caution is necessary when estimating their pathogenicity using this model.

This study has certain limitations that should be considered. In cases where patients exhibit multi-allelic mutations, it is important to ascertain the presence and abundance of a mutation in the gene locus. Precise quantification of *EGFR* mutation abundance has been reported to not only enable better patient selection for EGFR-TKI treatment but

**Table 2  The clinical characteristic of probands with *RPE65* variants.**

| Patients | Age(years)/gender | BCVA LogMAR R/L | Illness duration (years) | Fundus | ERG | Others | Diagnosis |
|---|---|---|---|---|---|---|---|
| F1 | 10/F | 0/0 | 10 | WYD | undetectable dark-adapted | Nb | RP |
| F2 | 6/M | 0.60/0.92 | 6 | WYD, BD | Extinct | Nb | LCA |
| F3 | 6/M | 0.70/0.70 | NA | None | Extinct | None | LCA |
| F4 | 15/M | 0.52/0.52 | 15 | WYD, BD | Extinct | Nb | LCA |
| F5 | 8/M | NA | 8 | WYD | Attenuate | Nb | LCA |
| F6 | 9/M | 0.60/0.82 | 5 | None | Attenuate | Nb | RP |
| F7 | 28/M | 1/3.2 | 28 | BD | Extinct | Nb | RP |
| F8 | 6/W | 1/1.30 | 6 | WYD | Extinct | Nb | LCA |
| F9 | 10/M | 0.52/0.40 | 10 | WYD | Attenuate | Nb | LCA |
| F10 | 48/W | 3.20/3.20 | 48 | BD | Extinct | Nb; Ny | LCA |
| F11 | 20/M | 0.82/1 | 15 | None | Attenuate | Nb; Ny | LCA |
| F12 | 30/M | 2.90/2.90 | 30 | NA | Extinct | Nb; Ny | LCA |
| F13 | 30/W | 0.30/0.40 | 30 | BD | Attenuate | Nb | RP |
| F14 | 34/W | NA | 34 | BD | Attenuate | Nb | RP |
| F15 | 61/M | 3.20/3.20 | 61 | BD | NA | Nb | LCA |
| F16 | 25/M | 0/0 | 25 | None | Attenuate | Nb | RP |
| F17 | 7/M | 0.22/0.30 | 4 | None | Attenuate | Nb | RP |
| F18 | 52/M | 0/0.22 | 0.5 | BD | Attenuate | None | RP |
| F19 | 29/M | 2.9/2.9 | 7 | BD | Extinct | Nb | RP |
| F20 | 47/W | 3.20/3.20 | 47 | BD | Attenuate | Nb | RP |
| F21 | 66/W | 1.30/1.30 | 66 | WYD, BD | Extinct | Nb | LCA |

**Notes.**

F, Family; M, man; W, women; NA, missing value; Nb, night blindness; Ny, nystagmus; R, right eye; L, left eye; M, male; F, female; BCVA, best corrected visual acuity; LP (light perception), 3 LogMAR (3.2); HM (Hand Motion), 2 LogMAR (2.9); WYD, white or white-yellow dots; PD, Bone-spicule-like pigment; deposits; RP, retinitis pigmentosa; LCA, Leber congenital amaurosis.

**Table 3  The correlation between clinical characteristics and structural biochemistry in *RPE65* missense variants.**

|  | AA change | Helix | Helices | β-hairpins |
|---|---|---|---|---|
| BCVA | 0.515[**] | 0.234 | 0.234 | 0.330[*] |
| illness duration | 0.156 | 0.187 | 0.187 | −0.062 |
| BCVA/illness duration | 0.340[*] | 0.145 | 0.145 | 0.186 |
| Fundus | 0.487[**] | 0.426[**] | 0.426[**] | 0.152 |
| ERG | 0.387[*] | 0.361[*] | 0.361[*] | 0.151 |

**Notes.**

Kendall's tau b correlation analysis.

[*]Correlation is significant at the 0.05 level (2-tailed).

[**]Correlation is significant at the 0.01 level (2-tailed).

also to facilitate the development of more effective treatment strategies for patients with a low abundance of EGFR mutations (*Zhou et al., 2011*). In our current model, the focus is limited to the protein structure at the mutation site, as information regarding mutation

abundance is unavailable. This limitation may be a contributing factor to the suboptimal predictive accuracy observed in our study.

## CONCLUSION

In this study, we investigated the relationship between pathogenic and population missense variations of RPE65 and protein 3D features and developed a novel prediction model (AUC = 0.7531). Furthermore, we analyzed the correlation between phenotype and protein structure in a Chinese cohort of patients with *RPE65* missense variants. We developed a complementary method presenting a novel approach to predicting the potential pathogenicity of *RPE65* missense variants based on protein structure. Our findings may provide valuable insights for the accurate diagnosis of *RPE65*-mutated inherited retinal diseases.

## ACKNOWLEDGEMENTS

We thank all the researchers who contributed to the detection of *RPE65* variants.

### Funding

This work supported by the Program of Shanghai Academic Research Leader (20XD1401100). Aging and women's and children's health Special project of Shanghai Municipal Health Commission, 2020YJZX0102. Shanghai Municipal Science and Technology Major Projects (2018SHZDZX05). National Natural Science Foundation of China (NSFC) 81790641; 82271085. The funders had no role in study design, data collection and analysis, decision to publish, or preparation of the manuscript.

### Grant Disclosures

The following grant information was disclosed by the authors:
Program of Shanghai Academic Research Leader: 20XD1401100.
Aging and women's and children's health Special project of Shanghai Municipal Health Commission: 2020YJZX0102.
Shanghai Municipal Science and Technology Major Projects: 2018SHZDZX05.
National Natural Science Foundation of China (NSFC): 81790641, 82271085.

### Competing Interests

The authors declare there are no competing interests.

### Author Contributions

- Jiawen Wu conceived and designed the experiments, performed the experiments, analyzed the data, prepared figures and/or tables, authored or reviewed drafts of the article, and approved the final draft.
- Zhongmou Sun conceived and designed the experiments, performed the experiments, authored or reviewed drafts of the article, and approved the final draft.

- Dao wei Zhang performed the experiments, analyzed the data, prepared figures and/or tables, authored or reviewed drafts of the article, and approved the final draft.
- Hong-Li Liu performed the experiments, prepared figures and/or tables, and approved the final draft.
- Ting Li performed the experiments, prepared figures and/or tables, and approved the final draft.
- Shenghai Zhang conceived and designed the experiments, authored or reviewed drafts of the article, and approved the final draft.
- Jihong Wu conceived and designed the experiments, authored or reviewed drafts of the article, and approved the final draft.

## Human Ethics

The following information was supplied relating to ethical approvals (*i.e.*, approving body and any reference numbers):

Eye Institute, Eye and ENT Hospital, College of Medicine, Fudan University.

## DNA Deposition

The following information was supplied regarding the deposition of DNA sequences:

The transcription of RPE65 sequence was downloaded from GenBank https://www.ncbi.nlm.nih.gov/gene/6121.

## Data Availability

The code for developing a prediction model can be found in the Supplemental Information.

## Supplemental Information

Supplemental information for this article can be found online at http://dx.doi.org/10.7717/peerj.15702#supplemental-information.

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
