# Peer review of "Development of a novel prediction model based on protein structure for identifying RPE65-associated inherited retinal disease (IRDs) of missense variants"

_PeerJ, doi:10.7717/peerj.15702_

## Round 0.1 · original submission · Major Revisions

Please address the concerns of all reviewers and revise the manuscript accordingly.

Reviewer 1 ·

Basic reporting

1) First sentence of the results summary (13 mutations of RPE65 were first reported and the structure of frameshift and nonsense was prematurely terminated) is not clear – please clarify.
There are some spelling mistakes throughout the manuscript that need to be corrected, e.g. line 229 (combing)

2) The authors do not clearly articulate why it is important to differentiate between RP and LCA. A more important thing to distinguish would be pathogenic from non-pathogenic mutations as this would help determine whether a patient is a candidate for gene therapy. It is strange that the authors did not even mention gene therapy with Luxturna in the introduction or discussion.

3) Prior important studies on this topic have not been cited: These include “Predicting the pathogenicity of RPE65 mutations”, “In silico Mapping of Protein Unfolding Mutations for Inherited Disease” and “Carrier frequency and incidence estimation of RPE65-associated inherited retinal diseases in East Asian population by population database-based analysis.” It would be informative to comment on how the approach in the present study compares to these prior investigations.

4) R code is not annotated, which limits its usability by other researchers – add explanatory comments

Experimental design

The authors need to describe the diagnostic criteria for LCA vs RP in order for the results to be readily applied by other researchers and clinicians.

The authors do many different comparisons with the same dataset using P < 0.05 as the significance level. Does the P value need to be adjusted down to account for multiple comparisons?
An important factor that seems not to have been considered is the degree of conservation of the amino acid substitution, assessed for example using a substitution matrix like BLOSUM. Why did the authors not include this in the model?

Validity of the findings

c.93A>G Thr31Thr Missense Novel – this is a synonymous mutation, not missense. The authors need to check for errors and correct as needed.

C112S is not novel: It was previously reported by this research group in “Frequency and phenotypic characteristics of RPE65 mutations in the Chinese population” and needs to be cited as such.

Additional comments

This study by Jiawen Wu and colleagues focuses on the development of an algorithm that allows RPE65-associated retinitis pigmentosa (RP) and Leber congenital amaurosis (LCA) to be differentiated based on the location of the mutation within the structure and its associated secondary structure. The study adds to our knowledge of RPE65-associated mutations and has a strength in its explicit use of RPE65 structural information to make predictions. The study would be strengthened by the inclusion of biochemical assays to assess the functional impact of mutations on RPE65 activity. However, I understand these methods might not be accessible to the research group. My suggestions for the manuscript are as follows:

Reviewer 2 ·

Basic reporting

The authors have reported a structure-function relationship of RPE65, an isomerohydrolase, mutations in which lead to inherited retinal degeneration. Through previously reported as well as 21 new patient mutations, relationships are drawn between the predicted structures of RPE65 mutants and two rare genetic disorders: retinitis pigmentosa (RP) and Leber congenital amaurosis (LCA). The paper explains the need for this study, includes details of their methods, results, and limitations. It is written intelligibly, includes figures and tables relevant to their claims as well as references to previous studies of this nature. Necessary approvals have been taken for using patient data. The authors also share their R code, which is appreciated. However, the captions of the supplementary figures are missing.

Experimental design

Major points
1. The main objective of this paper is to design a classifier that can classify a patient as having RP or LCA based on the mutations in their RPE65 genes (when no other confounding genes are mutated). While they mention that precision diagnosis of RPE65-related diseases is difficult, RP and LCA occur at different stages of life—LCA in the first year of life and RP in late childhood/early adulthood. Hence, the need for this classifier is not sufficiently established. The authors need to detail how such a classifier will help with predicting prognosis or treatment compared to what happens currently.
2. All conclusions about structure are drawn using the AlphaFold model of human RPE65. The model is a monomer, but crystal structures of cow RPE65 (~99% identical sequence) reveal that it functions as a membrane-associated dimer. Any reasonable analysis would need to consider this as the starting structure, not the AlphaFold monomer.
3. How they are incorporating the structural models for the mutant/truncated proteins predicted using SWISS-MODEL is unclear. Also, homology modeling programs are notoriously bad at predicting the structural effect of point mutations (as is AlphaFold in many cases).
4. The authors conflate test set with training set. They did the LASSO analysis on 208 mutations pulled from the HGMD database for feature selection. How is this a test set?
5. The majority of patients have heterogeneous alleles, i.e. each copy of RPE65 has different mutations. How their classifier works based on looking at structures of just one copy is unclear.
6. What about benign mutations, how does that factor into the model? The authors have not discussed that at all.
7. All mutations at one site are not the same: missense mutations depend on which residue gets mutated to which residue; frameshifts depend on where they occur and to which frame the protein gets shifted. By just classifying based on which point the mutations occur at rather than considering the type of mutation and what effect it has on the protein stability, interactions etc., the authors are oversimplifying the problem.
8. In Supplementary Table 1, it appears that not all patients have unambiguous clinical diagnoses. Six patients have a “?” next to their diagnosis and it seems like in two of the case it is not even clear if it is LCA or RP.

Minor Points
1. Line 98: “Optic coherence tomography (OCT) data was missing a lot” needs to be changed to “Optic coherence tomography (OCT) data was missing for several patients”
2. Line 150: Please rephrase: “The correlation of the other phenotypes was not correlated with illness duration”

Validity of the findings

1. With a very limited data set, the authors try to force correlations by feature selection. RPE65 is a highly conserved protein in mammals. It is expected that several mutations will cause pathogenic effects, but there could be different outcomes of each mutation: altered stability, fold disruption, dimer disruption, membrane-association disruption, active site disruption etc. The authors cite a paper where secondary structural features have been used as predictors for pathogenicity, but that paper explains why helices in BRCA1 are more prone to pathogenic missense mutations: because it binds DNA with those helices. The authors of this article make no such attempt to explain their results in any meaningful structural way.
2. With a dimer structure already available, it would be more meaningful to map the missense patient mutations—both new ones reported in this study and prior ones—to that structure and try to form a causative model of why LCA happens earlier in life than RP. Are the mutations in LCA more disruptive than those in RP?
3. To calculate the accuracy of any method, the ground truth in the training/validation/test set needs to be clearly established. Supplementary table 1 suggests that the diagnoses of more than a quarter of the patients was ambiguous. Calculating any accuracy metric with this vagueness is will only cause misleading results.

Due to the shortcomings in experimental design as well as the validity of the results, this article is fundamentally flawed. In its current state, no specific suggestions could improve it to a point of acceptability.

Additional comments

In their effort to classify patients as having RP or LCA, the authors miss commenting on a rather important point that others in literature have mentioned: perhaps they are the same disease which exists on a spectrum with LCA being the more aggressive form and RP being a milder form. Perhaps the reason why clinicians could not distinguish between the two in 6 out of 21 patients is because those cases were in the middle of this spectrum.

·

Basic reporting

The idea of using protein secondary structure to predict phenotype seems like a good approach. However the motivation of the study is not clear. Why are the authors interested in distinguishing the RP vs LCA IRD..

Please further describe the raw data table column names ? Eg what is ACMG category or BCVA or logMar

The figure and table captions need to be more descriptive. In Fig2, what does the color represent ? Fig2B the mutation labels are not readable. What does the grey color represent in Fig 2C-H.

 “Although it was known that secondary structure affected protein function, few studies searched for the correlations between secondary structure and phenotypes in RPE65 mediated IRDs. “ Please provide appropriate reference here

The supplementary figures and tables do not have any captions ? It is difficult to interpret the supplementary materials because of that

Table S1 has spelling errors in column names.  (Dimer binding residues ). Also some of the column names need more description such as how is “active site cavity” defined ? What are helix-helix interacs ?


line 163: “RPE65 variants affect the protein structure and function”. This needs to be supported with appropriate references in the literature

Line 164 “A correlation analysis between ….” The sentence needs to be corrected. The correlation analysis is performed between protein secondary structure and phenotype and not protein structure and phenotype

Are all the 208 mutations obtained from HGMD are missense mutations ?

Experimental design

The motivation of the study seems to be to predict if the mutation would cause LCA and RP. But it is not clear why ? Do the two diseases have differences in the treatment used for them ?



It is not clear in results section 3.1 that why "classifying RP and LCA is important but difficult" ?

It is not clear why the author predicted the structure of the RPE65 with missense mutations ? If the protein structure is stable ,  SWISS-MODEL won’t be able to capture structure changes due to single mutations anyway..

How many residues does RPE65 has ? This should be mentioned at least in line 160 (encoding less than 533 amino acids). If all the insertion and deletion mutations are leading to premature termination, I do not understand why the authors predicted the tertiary structure of them using SWISS-MODEL.

Validity of the findings

It is appreciated that the authors acknowledge the major limitations of the study:
-small validation dataset
-weak accuracy (C-index-0.657)
-lack of experimental verification

Can the author comment on the similarity/differences in the known crystal structure of RPE65 and the structure they predict using alphafold ?

Can the author speculate on how the secondary structure of the mutant is possibly affecting the type of disease it causes (LCA vs RP in this case) ?

---

## Round 0.2 · Major Revisions

Please address the remaining issues indicated by the reviewer and amend the manuscript accordingly.

Reviewer 1 ·

Basic reporting

no comment

Experimental design

The authors have changed the focus of their study to differentiate between pathogenic and non-pathogenic missense mutations, which is a more interesting research question compared to the one being investigated in the previous version. The authors consider basic structural information about RPE65 in their predictive model such as secondary structure elements, active site involvement, etc. Their method is a little confusing because it seems like they include three different levels of granularity in the analysis and use all three when describing the risk level of the substitution. For example, in lines 192-196, it is confusing when the authors describe both "sheet" and "beta sheet" as being correlated with IRD-associated missense mutations. These are the same structural feature so it is strange they are being mentioned twice. Going back to a comment from my first review, it seems like the authors a key piece of information that would boost the predictive power of their model and this is how conserved the particular missense substitution is. This is information that could have been easily obtained from a standard substitution matrix or from looking at RPE65 ortholog sequence alignments or sequence logos. Ignoring this key information is a major shortcoming of the revised study in my view.

Validity of the findings

The approach taken by the authors to use structural information in a predictive model is a significant advance over the approaches used in the past for predicting pathogenicity. It is interesting that relatively simple classifications based on secondary structure features led to the development of model with reasonable predictive power.

Additional comments

I still see several typos in the revised manuscript that need to be corrected. Examples include:

helix-helix interacs (OR = 0.43, P = 0.046) - spell out interactions
located in the sheet of 3-class-secondary - does not make sense
and population of RPE65 missense - needs to be fixed
that the missense located in these sites is prone to be pathogenic - missense is repeatedly used without "mutation" afterward.
easily missing factors - needs to be fixed
Fig. 1 - "embrane" - needs to be fixed

These are just a few examples and there are many others. The paper needs to be carefully edited, if possible by a fluent English speaker.

·

Basic reporting

no comment

Experimental design

no comment

Validity of the findings

no comment

Additional comments

The authors have provided satisfactory response

---

## Round 0.3 · accepted · Accept

All remaining concerns are adequately addressed. The reviewer is satisfied with the revision. The amended version is acceptable now.

Reviewer 1 ·

Basic reporting

The authors have addressed my prior concerns. The current version is a valuable contribution to the RPE65 literature.

Experimental design

The authors included appropriate factors in their analysis

Validity of the findings

No issues